



# Contrasting strategies of nutrient demand and use between savanna and forest ecosystems in a Neotropical transition zone

Marina Corrêa Scalon[1,2], Imma Oliveras Menor[1,3], Renata Freitag[4], Karine S. Peixoto[4], Sami W. Rifai[1,5], Beatriz Schwantes Marimon[3], Ben Hur Marimon[3], Yadvinder Malhi[1]

[1]Environmental Change Institute, School of Geography and the Environment, University of Oxford, Oxford OX1 3QY, UK
[2]Programa de Pós-graduação em Ecologia e Conservação, Universidade Federal do Paraná, Curitiba, PR, 81531-990, Brazil
[3]AMAP (botAnique et Modélisation de l'Architecture des Plantes et des Végétations), Univ. de Montpellier, CIRAD, CNRS, INRAE, IRD, Montpellier Cedex 5, France
[4]Programa de Pós-graduação em Ecologia e Conservação, Universidade do Estado de Mato Grosso – UNEMAT, Nova Xavantina, MT, 78690-000, Brazil
[5]ARC Centre of Excellence for Climate Extremes, Sydney, NSW, Australia

*Correspondence to*: Marina C. Scalon (marina_scalon@yahoo.com.br)

**Abstract.** The total demand and uptake of nutrients by vegetation is rarely quantified or compared across vegetation types. Here, we describe different nutrient use and allocation strategies in Neotropical savanna (cerrado) and transitional forest (cerradão) sites, report leaf nutrient resorption and calculate ecosystem-level nutrient use efficiency. For the first time, we couple net primary productivity (NPP) estimates with nutrient stoichiometry to quantify nutrient demand and nutrient flows at the whole stand scale for different components of vegetation biomass. The two vegetation types showed similar mean nutrient concentration and nutrient resorption efficiency except for wood P concentration that was 4-fold higher in cerrado than cerradão species. The cerradão showed higher canopy NPP, while fine roots and wood NPP were similar for the two vegetation types. Nutrient requirement in the two vegetation types was dominated by the demands of the canopy, with canopy resorption contributing generally more than 50% of the total canopy demand for nutrients, while less than 35% of N, P, K, Ca and Mg were allocated to wood or fine roots. Proportionally, the savanna site showed higher nutrient demand from fine-roots (over 35% of total nutrient demand) and for the wood component (over 13% of total nutrient demand), while ~60-70% of cerradão nutrient demand was allocated to the canopy. The proportional difference in nutrient allocation to the different biomass components suggesting cerrado species are more efficient in fine root production, but less efficient in producing wood. Our findings suggest that cerradão species are limited in P and K, inducing a higher resorption to compensate for low uptake. Moreover, we found that N uptake for cerradão was higher with lower N use efficiency, leading higher N demand compared to the cerrado. This trade-off explains how similar soils and the same climate dominated by savanna vegetation can also support forest-like formations. The lack of difference in Ca and Mg use and uptake efficiency also suggests these ecosystems are able to acquire all Ca and Mg they need. Tree species composition is likely the major factor regulating nutrient use, limiting vegetation transitions and influencing nutrient demand at landscape scales.



# 1 Introduction

Net primary productivity (NPP), together with carbon and nutrient cycling are important ecosystem processes that may be primarily limited by nutrient availability (Aragão et al., 2009; Quesada et al., 2009; Fernández-Martínez et al., 2014). Even

though there is a major interest in the relationship between nutrient and carbon for predictive models on how tropical ecosystems will respond to increasing atmospheric CO2 and climate change (Malhi, 2012), the influence of nutrient availability in carbon balance and sequestration is not well understood. Most studies of nutrient demand are based on simple concentrations and stocks of nutrients in, for example, leaves, wood and soil. In order to assess the *flow and demand* of nutrients in vegetation, it is necessary to couple the stoichiometry of key plant organs with the rates of production of those

organs, which are assessed through measurements of NPP. In the context of tropical ecosystems, the availability of forest sub-component NPP data has increased substantially in recent years, most notably through the activity of the Global Ecosystems Monitoring (GEM) network (Malhi et al., 2020). The coupling of NPP and stoichiometry data opens the potential for detailed description of plant nutrient demand and nutrient cycling in a variety of ecosystems. Here we demonstrate how such coupling of NPP estimates with nutrient stoichiometry can be conducted in the context of Neotropical

savanna and transitional vegetation types.

Although climatic fluctuations have major role shaping the forest and savannas occurrence and dynamics (e.g. expansions and retractions) globally, mesic and humid savannas normally persist over areas where climatic conditions could support forest formations (Cole, 1986; Solbrig, 1996). Therefore, in addition to climatic conditions, nutrient availability and species shifts are expected to be important environmental filters (Lehmann et al., 2011). It is suggested that nutrients may be the

primary determinant of tree biomass worldwide, since trees experiencing higher nutrient availability can potentially have faster growth rates and therefore be more competitive against grasses and avoid the grass fire trap (Hoffmann et al., 2012). Key limiting nutrients are usually presumed to be nitrogen or phosphorus (Reich and Schoettle, 1988; Reich et al., 2009), but potassium (K), calcium (Ca), and magnesium (Mg) may also play a role (Chapin III, 1980; Grime, 2006). Savanna soils tend to be highly deficient in multiple nutrients (Bucci et al., 2006; Haridasan, 2008; Bustamante et al., 2012) and some authors

attribute soil fertility as the main factor driving abrupt transitions between savannas and forests (Bond, 2010; Wang et al., 2010; Viani et al., 2011). In the Neotropical mesic savannas, when fire is excluded, gradual forest expansion occurs by encroachment of trees, forming forests-like vegetation (Durigan and Ratter, 2006; Hoffmann et al., 2012; Rosan et al., 2019), meaning that nutrient availability alone is also not the ultimate constraint on forest formation (Pellegrini, 2016). However, nutrients might influence vegetation encroachment processes by either restricting tree growth rates, maintaining a grass layer

that facilitates fire; or constraining the ability of forests to form regardless of fire (Case and Staver, 2017).

Soil nutrient availability is determined not only by soil properties, but also by water availability and vegetation structure and composition. For instance, high biomass results in high nutrient turnover and availability (Nardoto et al., 2006; Pellegrini, 2016), and species-specific leaf functional traits can affect demand and turnover of nutrients (Craine et al., 2008). While





savanna-like vegetation, such as the typical cerrado is defined by shrubs and trees distributed in a continuous grass and
herbaceous layer, savanna-forest transitional vegetation, such as the cerradão is composed by an almost continuous canopy
of trees (usually 70-80% canopy cover) with a thin herbaceous understory layer (Oliveras and Malhi, 2016). Few plant
species are shared between cerrado and cerradão, implying a distinct set of traits and ecological strategies (Hoffmann et al.,
2003; Marimon-Junior and Haridasan, 2005; Marimon et al., 2006). There is an expected evolutionary trade-off involved in
slow and fast growing species in relation to high-nutrient and low-nutrient habitats, in which plants adapted to nutrient-poor
habitats should be more efficient (i.e., build more organic matter for a given unit of nutrient required) than fast growing
plants from nutrient-rich environments (Vitousek, 1982; Vitousek, 1984; Chapin III et al., 1986). Indeed, higher resorption
rates are expected in low-nutrient environments (Aerts and Chapin, 1999; Wright and Westoby, 2003; Pellegrini, 2016).
Leaf nutrient concentration and leaf nutrient resorption are obviously major aspects that influence nutrient cycling in a plant
community. However, nutrient allocation to different organs (e.g., leaves, roots, wood, and bark) can also have important
effects in nutrient use and supply in the ecosystem, since different species can have distinct demands for nutrients in terms of
tissue stoichiometries. In addition, nutrient deposition and mineralization from different plant material will directly influence
nutrient availability (Wardle et al., 2004). While wood can provide a well-defended and long-lived storage organ for
nutrients, water and carbohydrates (Chapin III et al., 1990), leaves and fine roots are short-lived and much more vulnerable
to herbivores, especially under high nutrient concentration (Coley et al., 1985; Moles et al., 2013).
To effectively evaluate nutrient use in forest-savanna transitions, there is the need to quantify the nutrient requirement of
each vegetation type, which should consider differences in biomass investment for the distinct components (e.g., canopy,
wood, roots). By calculating nutrient demand in the savanna-forest transition using literature proxy equations and data,
Pellegrini (2016) found that populations of Brazilian savanna species, compared to forest species, required double the N and
P to form closed canopies, but the N and P annual demand was substantially greater for forest tree species. These findings
suggest that even with lower nutrient requirements to form a forest, transitional forest species may experience higher
limitations due to their higher nutrient demand. Species composition and vegetation structure are very distinct between the
cerrado and cerradão vegetation (de Oliveira et al., 2017), and the mechanism by which these differences occur is still poorly
understood, but may involve trade-offs between nutrient requirements and adaptations to fire (Silva et al., 2013), and
vegetation encroachment by nucleation of keystone species, such as *Tachigali vulgaris* (Morandi et al., 2016). Here, we
describe different nutrient use and allocation strategies in savanna (cerrado) and transitional forest (cerradão) species, report
leaf nutrient resorption and calculate ecosystem-level nutrient use efficiency based on dynamic nutrient flows rather than
one-off static estimates of nutrient stocks. We set out to test the following questions and hypotheses:

(I)       How do leaf nutrient concentration and resorption vary across savanna-forest species? We expect that there will be
more nutrients in different plant organs in cerradão species compared to cerrado vegetation. Moreover, based on plant



strategies and resource availability limitations, we expect that cerrado species will show higher nutrient resorption efficiency.

(II)      What is the nutrient demand of the cerrado and cerradão sites? We expect that cerradão will have higher NPP for all components, and therefore, higher nutrient demand.

(III)     What is the partitioning of nutrient demand between canopy, wood and fine roots, and how much of this demand is
met by translocation of leaf nutrients rather than by new uptake? We expect that the canopy will dominate nutrient demand in both vegetation types, but that the majority of this demand will be met by nutrient translocation from senescing leaves.

## 2 Material and Methods

### 2.1 Field site

The study plots, part of the GEM intensive carbon plot network (Malhi et al., 2020), are located in a typical mosaic area of
cerrado and an adjacent cerradão vegetation in the Bacaba Municipal Park (BMP; 14◦42’22” S, 52◦21’07” W), Nova Xavantina, Mato Grosso State, Brazil. The park of approximately 500 ha is located in the transition between Cerrado and Amazon biomes; the climate is Köppen's Aw (i.e., tropical savanna climate), with a highly seasonal rainfall and a prolonged and intense dry period during May to September (Marimon-Junior and Haridasan, 2005).

Two 1-ha plots have been monitored every year since 2010, for vegetation dynamics following standard RAINFOR protocol
(Phillips et al., 2002) and are 100 m apart from each other. Tree identity, size, location, and growth data for each plot and census are curated and available under request at the ForestPlots.net database (Lopez-Gonzalez et al., 2011). Our sites represent long-term fire protected vegetation types, and a rare case of very well-preserved typical cerrado and cerradão vegetation in the Amazon and Cerrado transition (de Oliveira et al., 2017). Typical cerrado is a cerrado stricto sensu subtype with predominantly arboreal-shrubby vegetation, 20 to 50% of tree cover, and tree heights between three and six meters
(Ribeiro and Walter, 1998). The cerradão is considered an ecotonal community (Ratter et al., 1973), characterized by mostly continuous canopy and the abundance of species that indicate the transition between forests and savannas on the southern Amazonian border, such as Hirtella glandulosa and Emmotum nitens (Marimon et al., 2006).

Within each 1 ha plot, five soil cores to the depth of 200 cm were collected and soil samples were partitioned by eight depth layers (0-5, -10, -20, -30, -50, -100, -150, -200 cm), also following the RAINFOR protocol (see
http://www.rainfor.org/en/manuals). Despite slightly sandier soils in the cerrado, the soil in both areas are very similar, and characterized as dystrophic yellow oxisols, with low nutrient concentration, reduced sum of the bases, and low cation exchange capacity (Marimon-Junior and Haridasan, 2005). They are highly acidic, Alic, well drained and without concretions up to two meters deep (Table S1).





## 2.2 Nutrient sampling

In each area, we randomly selected five individuals of the seven most abundant species representing ca. 80% of the community basal area (Table S2). For each individual we collected samples from different tissues: heartwood and sapwood, inner and outer bark, branch, mature and old leaves, and fine roots. Inner and outer bark can be easily visually distinguished, but we also used magnified lenses to confirm our classification whenever there was any uncertainty. Ten fully expanded mature sun leaves were collected during the wet season (January 2018) and 10 recently senescent leaves were collected

during dry season (July 2018) because of the deciduousness of most species. We considered senescent leaves to be those that fall with a gentle flick to the branch (Wright and Westoby, 2003). Collected branches were approximately 1cm diameter without bark, which was removed before analysis. Wood core samples and bark were collected at breast height (DBH ≈ 1.30 m). Wood samples were extracted using a 4.3 mm Haglof increment borer to a depth of half the DBH of the tree.

Fine root samples were collected at 20 cm depth soil near the coarse root of each individual. While this might include roots

from outside the target individual, we were interested in root nutrient composition at the community scale, therefore this approach was sufficient to address our key question. Samples were sieved and washed with tap water. After being oven dried at 70 °C until constant weight, dried samples were sent to Laboratório de Análises de Viçosa-MG, for nutrient concentration analyses. N and P (g kg – 1) were determined by Kjedahl digestion and UV-Vis spectroscopy, respectively, and the other nutrients (Ca, Mg and K, g kg – 1) were determined by atomic absorption spectrometry.

Nutrient resorption efficiency (RE) was calculated following Eq. (1):

$$RE = \left(1 - \frac{Nut_{old}}{Nut_{mat}} MLCF\right) \times 100 \tag{1}$$

where $Nut_{mat}$ is the nutrient concentration of mature leaves and $Nut_{old}$ is the nutrient concentration of senesced leaves and $MLCF$ is the green to senesced leaf Ca ratio (Vergutz et al., 2012). This calculation is based on the assumption that Ca is not translocated, and enables correction for differences in leaf mass per area between mature and senescent leaves. Community-

weighted means for each nutrient were calculated using the proportional basal area for each species.

## 2.3 Estimation of NPP

Net primary productivity is the rate of organic material production, and can be quantified at the scale of an individual plant or an ecosystem (Malhi et al., 2011). Here we quantified the major components of NPP, including the canopy (leaves, twigs and reproductive parts), wood (stem, coarse roots and branches), and fine roots. Methods are described in detail in Peixoto et

al. (2017), Peixoto et al. (2018) and Malhi et al. (2015).

For stoichiometric calculations, carbon was assumed to be 50% of the biomass for wood and canopy components, 48% for fine roots and 39% for litter (Biot et al., 1998; Paiva et al., 2011). Nutrient uptake (kg ha$^{-1}$ year$^{-1}$) and nutrient demand (kg ha$^{-1}$ year$^{-1}$), were calculated following Eq. (2) and Eq. (3), respectively:



$$Nut_{uptake} = \left( \left( \frac{NPP_{canopy}}{C:Nut_{mat}} \right)(1 - RE) \right) + \left( \frac{NPP_{wood}}{C:Nut_{wood}} \right) + \left( \frac{NPP_{fine\ roots}}{C:Nut_{fine\ roots}} \right) \tag{2}$$

$$Nut_{demand} = \left( \left( \frac{NPP_{canopy}}{C:Nut_{mat}} \right) \right) + \left( \frac{NPP_{wood}}{C:Nut_{wood}} \right) + \left( \frac{NPP_{fine\ roots}}{C:Nut_{fine\ roots}} \right) \tag{3}$$


where NPP is plot-mean NPP (Mg C ha$^{-1}$ year$^{-1}$), and C:Nut is plot-mean nutrient stoichiometry.

Thus, we calculated nutrient uptake efficiency (kg C per kg nutrient) as the ratio of total NPP (NPP$_{canopy}$ + NPP$_{wood}$ + NPP$_{fine}$
$_{roots}$) and Nut$_{uptake}$; and nutrient use efficiency (kg C per kg nutrient) as the ratio of total NPP and Nut$_{demand}$. Hence nutrient
uptake efficiency takes into account that a large part of nutrient demand can be met by translocation from senescing leaves
and therefore does not require uptake from the soil.

## 2.4 Statistical analyses

To compare community-weighted nutrient concentration average means and resorption efficiencies between the two
vegetation types and between different organs, we performed a multivariate analysis of variance (MANOVA) followed by
univariate analysis of variances (ANOVA's) and Tukey HSD post-hoc test. Data normality and homogeneity of variances
assumptions were previously checked with Shapiro-Wilk multivariate normality test using package 'mvnormtest' (Jarek and
Jarek, 2009) and Levene test using package 'car' (Fox et al., 2012), respectively. All statistical analyses were performed in R
software version 4.0.1 (R Core Team, 2019).

To account for the systematic uncertainties intrinsic to the methods used to NPP calculations for the different components,
we assigned sampling uncertainties to each measurement and propagated them through our calculations. Following the
standard rules of quadrature (Hughes and Hase, 2010), we calculated the standard error of the mean (SE), and propagated it
by adding the components in quadrature, assuming the uncertainties in the independent variables were uncorrelated. To
calculate differences between sites for nutrient use and uptake efficiencies, we calculated z-score using the average value for
each site and the propagated error.

## 3 Results

### 3.1 Savanna-forest differences

For species mean comparisons, transition forest species showed higher leaf (green and old leaves), branch and root N
concentration, while savanna species showed higher wood nutrients in general (higher N, P, K) for both sapwood and
heartwood (Fig. S1). Cerrado species also showed higher leaf Mg concentrations and higher P and K inner bark
concentrations. For community-weighted means, however, we did not find an identifiable interaction effect between the
different plant organs and vegetation types (MANOVA: F = 1.28, P = 0.134), meaning that generally there was no difference



in nutrient concentration between cerrado and cerradão across the different tree components when species abundance and basal area were taken into account (Fig. 1). Moreover, despite slightly higher nutrient resorption in the savanna, there were no differences between the two vegetation types (MANOVA: F = 0.23, P = 0.916; Fig. 2A). In general, we found higher P resorption (cerrado: 58.50 ± 18.97% and cerradão: 50.44 ± 15.72%) than N (cerrado: 49.46 ± 19.44% and cerradão: 39.58 ±

15.39%) for both sites and all species (Fig. 2).

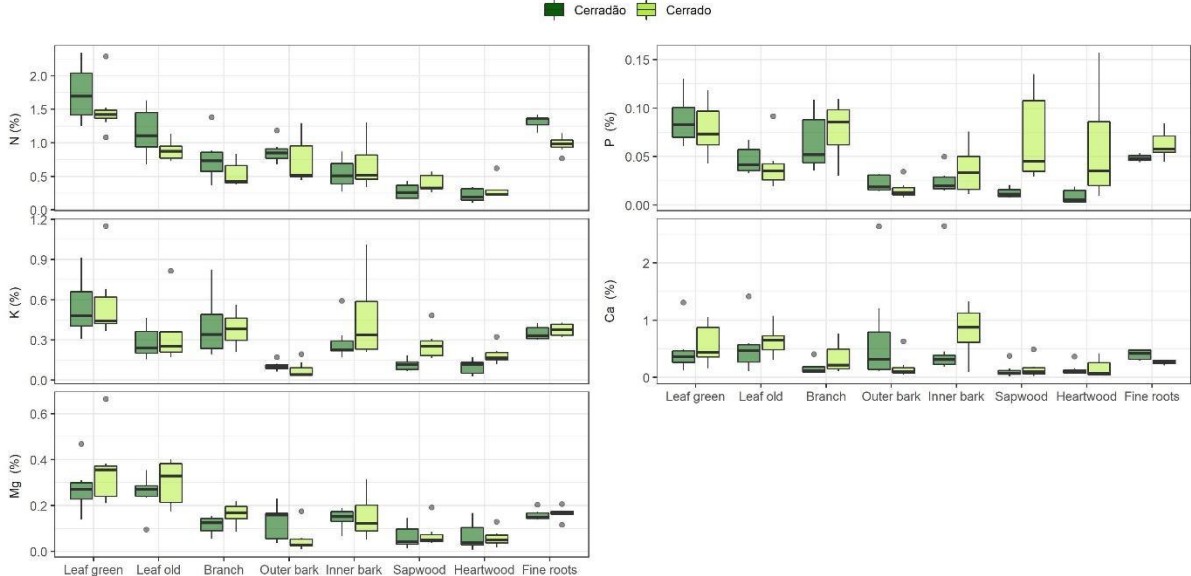

**Figure 1: Macronutrient (N, P, K, Ca and Mg) concentrations (%) for different plant organs (leaf, branch, outer bark, inner bark, sapwood, heartwood and fine roots) in cerradão (dark green boxes) and cerrado (light green boxes) vegetation. The continuous line within the box shows the median, and error bars show 10 and 90 percentiles (n = 7**
**species per vegetation type). Asterisks (\*) represent differences between the two ecosystems (P < 0.05).**





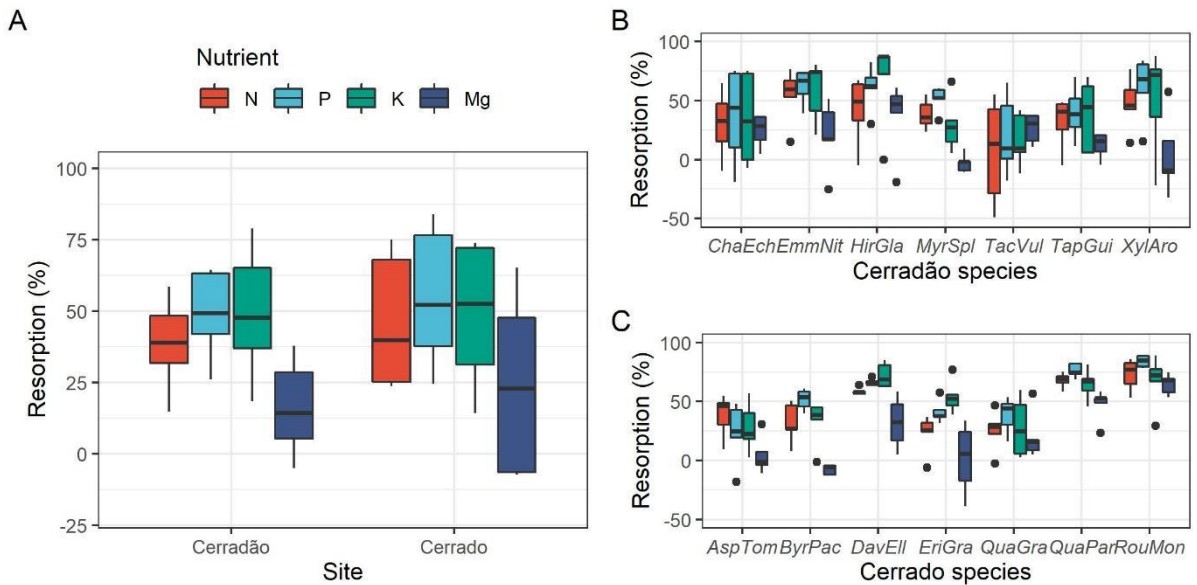

**Figure 2: Nutrient (N, P, K, Mg) resorption efficiency (%) averaged for both vegetation types considering the community weighted mean for each species (A) and calculated for each of the species representing the transition forest (cerradão, B), savanna (cerrado, C). The continuous line within the box shows the median, and error bars show 10 and 90 percentiles (n = 5 individuals per species). Species abbreviation refer to the first three letters of the genera followed by the first three letters of the specific epithet, as follow from left to right: cerradão species (*Chaetocarpus echinocarpus*, *Emmotum nitens*, *Hirtella glandulosa*, *Myrcia splendens*, *Tachigali vulgaris*, *Tapirira guianensis*, *Xylopia aromatica*); and savanna species (*Aspidosperma tomentosum*, *Byrsonima pachyphylla*, *Davilla elliptica*, *Eriotheca gracipiles*, *Qualea grandiflora*, *Qualea parviflora*, *Roupala montana*).**

**3.2 Nutrient demand and partition**

Cerradão yielded higher NPP for canopy (cerradão: 4.78 ± 0.12 and cerrado: 2.38 ± 0.08 Mg C ha-1 year-1) but similar NPP values for wood (cerradão: 2.77 ± 0.58 and cerrado: 2.79 ± 0.45 Mg C ha-1 year-1) and lower fine roots NPP (cerradão: 2.96 ± 0.76 and cerrado: 3.63 ± 0.81 Mg C ha-1 year-1; Fig. 3a) components. In agreement with our expectations, nutrient demand was generally higher for cerradão species for all nutrients except for nutrient demand for the wood component, especially for phosphorus, which was 4-fold higher in the cerrado site, but also for N, K, Ca and Mg (Fig. 3).

Biogeosciences Discussions Open Access

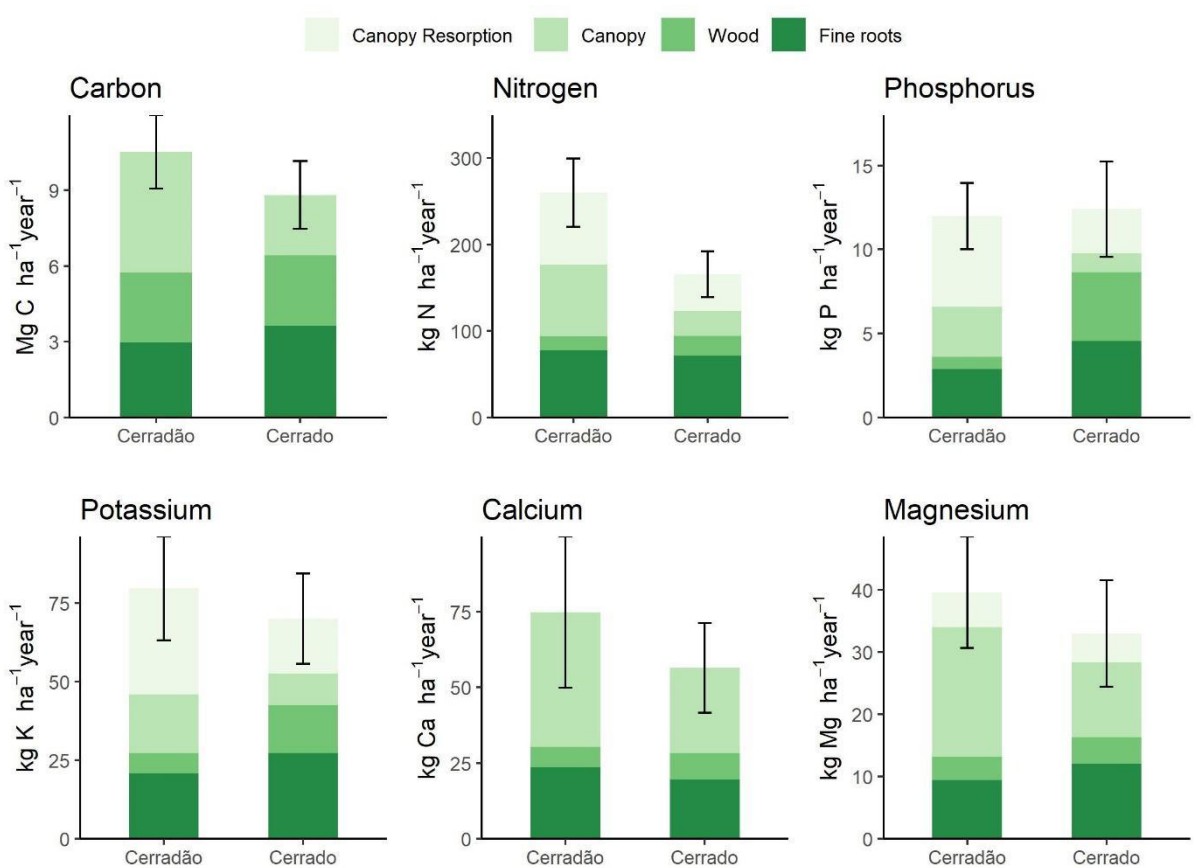

**Figure 3: Net primary productivity (NPP, Mg C ha-1 year-1), and nutrient demands (N, P, K, Ca and Mg) partitioned into biomass components: canopy demand met by resorption (canopy resorption), canopy demand met by new uptake (canopy), wood and fine roots in cerradão (transition forest) and typical cerrado (savanna). Error bars indicate ± SE for all components pulled together (see Table 1, Figure S2 for individual errors).**

Nutrient demand in the cerradão was predominantly allocated to canopy, with nutrient demand met by resorption corresponding to generally ~ 50% of the total canopy demand, while less than 35% of the N, P, K, Ca and Mg were allocated to the wood or fine roots (Table 1). For the cerrado, proportional allocation of nutrients was more equally partitioned across the different organs. For instance, in the cerrado, root nutrient demand was similar to total canopy nutrient demand for all nutrients, while P demand to the wood component corresponded to ~ 50% of the total phosphorus demand. In contrast, cerradão proportionally demanded more nutrients to the canopy, while cerrado proportionally demanded more N, P and K, Ca and Mg to the wood and fine roots (Table 1).



**Table 1.** The total (mean ± SE, kg ha$^{-1}$ year$^{-1}$) and proportional (%) partitioning of nutrient demand between canopy, wood and fine roots, together with the component of overall nutrient demand met via resorption from senescent leaves, for transitional forest (*cerradão*) and savanna (*cerrado*) sites.

| Nutrient | Cerradão | | | | Cerrado | | | |
| | Canopy | | Wood mean ± SE (%) | Fine root mean ± SE (%) | Canopy | | Wood mean ± SE (%) | Fine root mean ± SE (%) |
| | Total mean ± SE (%) | Resorption mean ± SE (%) | | | Total mean ± SE (%) | Resorption mean ± SE (%) | | |
|---|---|---|---|---|---|---|---|---|
| N | 166.52 ± 25.83 (63.98) | 83.09 ± 16.59 (49.90) | 15.93 ± 3.52 (6.12) | 77.83 ± 10.32 (29.90) | 71.29 ± 13.67 (43.00) | 42.35 ± 10.23 (59.41) | 23.51 ± 3.94 (14.18) | 70.99 ± 8.93 (42.82) |
| P | 8.37 ± 1.43 (69.97) | 5.41 ± 1.05 (64.55) | 0.73 ± 0.17 (6.11) | 2.86 ± 0.38 (23.93) | 3.75 ± 0.86 (30.25) | 2.63 ± 0.67 (70.02) | 4.11 ± 1.27 (33.14) | 4.53 ± 0.69 (36.60) |
| K | 52.29 ± 12.18 (65.68) | 33.64 ± 8.76 (64.35) | 6.55 ± 1.31 (8.22) | 20.77 ± 3.04 (26.10) | 27.42 ± 7.69 (39.17) | 17.38 ± 5.43 (63.40) | 6.55 ± 3.27 (21.87) | 27.27 ± 3.42 (38.96) |
| Ca | 44.42 ± 17.71 (59.33) | 0.00 | 6.81 ± 3.37 (9.09) | 23.64 ± 3.88 (31.58) | 28.07 ± 7.88 (49.71) | 0.00 | 8.85 ± 4.37 (15.68) | 19.54 ± 2.54 (34.61) |
| Mg | 26.37 ± 6.12 (66.63) | 5.55 ± 2.53 (21.05) | 3.80 ± 1.53 (9.60) | 9.41 ± 1.29 (23.77) | 16.65 ± 5.58 (50.48) | 4.61 ± 3.07 (27.68) | 4.28 ± 1.51 (12.98) | 12.05 ± 1.78 (36.54) |



P and K uptake efficiencies were ~45% and ~30% higher for the cerradão site respectively, while N uptake efficiency was ~17% higher in the cerrado site (Fig. 4). N, Ca and Mg uptake efficiencies were similar between cerradão and cerrado (Fig. 4). For nitrogen, cerradão uptake was higher with similar resorption factor and lower (~24%) N use efficiency, leading to higher N demand compared to the cerrado (Fig. 4). For P, there was a decrease in uptake in the cerradão coupled with an increase in resorption and 24% higher P use efficiency, leading to little overall change in P demand (Fig. 4). There was little difference in K uptake between the sites but the increased resorption in cerradão supports greater demand (Fig. 4). For Ca and Mg, uptake matched demand and there was little adjustment in resorption or nutrient use efficiency between both vegetation sites (Fig. 4).



**Figure 4: Nutrient uptake efficiency (kg C per kg [Nut]), nutrient use efficiency (kg C per kg [Nut]), resorption factor (i.e., the demand to uptake ratio), and the absolute value for nutrient demand (dark green bars) and uptake (light green bars) in cerradão (transition forest) and typical cerrado (savanna). Error bars indicate ± SE and asterisks show significant differences between sites (z-tests, * $0.01 < P < 0.05$; ** $0.001 < P < 0.01$; ***$P < 0.001$).**




## 4 Discussion

Unexpectedly, there was no major difference in cerradão and cerrado species nutrient content in the different plant organs. The only and most remarkable difference was in wood P concentration, where we found a clear trend of all cerrado species
allocating very high amounts of P to the inner and outer wood (Fig 1, Fig. S1). Wood nutrient allocation directly affects nutrient residence time at a whole tree level. Nutrients allocated to the canopy lead to larger flux via fast turnover meaning a shorter residence time, compared to nutrients allocated to wood biomass. Our results agree with the current literature suggesting that P in tropical ecosystems is a key limiting nutrient (Malhi et al., 2009; Quesada et al., 2010) and may be affecting transition ecosystem dynamics (Dionizio et al., 2018), suggesting two distinct strategies in the cerrado and cerradão
vegetation. In the cerradão, transition forest species may be allocating P to the canopy as an adaptive mechanism to maintain higher photosynthetic rates (Gleason et al., 2009; Reich et al., 2009) to support the high net primary productivity in low-P soils. In cerrado, savanna species might allocate P to the wood as a mechanism to extend P residence time in the living biomass (Aoyagi and Kitayama, 2016; Tsujii et al., 2020).

The strategy of storing P in wood biomass of cerrado species may be important to avoid P loss by recurrent fires. Although
soil P may even increase shortly after fire (Schaller et al., 2015), P-binding components in ash may decrease P availability, by forming low soluble P forms that are not readily available for plants (Schaller et al., 2015, Ngoc Nguyen, 2014 #8852). Furthermore, the loss of P stocks after fire because of volatilization remains anecdotal because P has a high volatilisation temperature (above 770C, DeBano, 2000) and Cerrado fires are usually very fast and do not achieve these high temperatures (Miranda et al., 1993). Although some experimental studies comparing the amount of nutrients in the fuel and in the ash after
fire found a ~50% reduction of P content (Pivello and Coutinho, 1992; Kauffman et al., 1994), Resende and others (2011) suggested that the decrease in organic-P fraction and P availability in cerrado burned plots compared to unburned plots was probably caused by the decrease of organic inputs through litterfall, rather than a direct effect of fire. In the same study area as ours, de Oliveira and others (2017) have found higher decomposition of the total biomass, and higher nutrients cycling through litterfall in the cerradão compared to cerrado site. Indeed, Mendes and others (2012) have shown that forest-like
vegetation types in the Cerrado biome (cerradão and gallery forests) presented higher microbial biomass and soil biological functioning compared to savannas. Therefore, we suggest that cerradão species may have adopted the strategy of relying on the soil biota activity and higher litterfall production to promote rapid nutrient cycling, whereas cerrado vegetation species may have evolved by natural selection to minimise P loss, by storing P in the wood. Nevertheless, further research in nutrient dynamics in these vegetation types is needed to test this hypothesis.
Nutrient resorption efficiency was also similar between cerrado and cerradão vegetation (Fig. 2), and our results are in agreement with the global average range for woody angiosperms (Vergutz et al., 2012). Our fine-scale leaf resorption results are consistent with previous reports suggesting that effective nutrient cycling in the cerradão compensates for the low saturation of exchangeable bases in the soil (de Oliveira et al., 2017). Moreover, the lower N uptake and use efficiency and the higher P and K uptake and use efficiency in the cerradão (Fig. 4) suggests that differences in tree community



270 composition and the species-specific functional traits may considerably influence nutrient cycling. For example, Tachigali vulgaris, which contributes to ~20% basal area in the cerradão (Table S2), was described as a key N-fixing species (de Castro et al., 1998) that promotes tree encroachment, by increasing enriched litterfall production and facilitating growth and development of other tree species (Morandi et al., 2016; de Oliveira et al., 2017). Indeed, the increased N uptake in the cerradão allows for a reduced NUE meaning more N in stoichiometry (Fig. 4).

275 Ultimately, cerradão species were more efficient in both P and K uptake and use (Fig. 4), which may explain how similar soils (Table S1) and the same climate can support forest-like formations. Cerradão vegetation require more N, Ca and Mg, and less P, but adjust the stoichiometry and boost P resorption, leading to similar overall usage of P (Fig.4), while K resorption is increased, which supports greater NPP despite similar net uptake (Fig.4). The relatively similar Ca and Mg uptake and use efficiencies between sites suggests that both ecosystems are able to satisfy their Ca and Mg requirements

280 (uptake tracks demand, with only modest adjustments in stoichiometry), whereas P, N and to some extent K, may be more limiting nutrients.

 In particular, the high P-use efficiency (PUE) exhibited for both cerradão and cerrado vegetation (Fig. 4), is an expected adaptation for dealing with low-P soils, and extending P residence time is an adaptive trait to increase PUE. Indeed, in Bornean and Australian low-P tropical forest species, P residence time in the canopy was shown to increase with P

285 deficiency (Gleason et al., 2009; Aoyagi and Kitayama, 2016; Tsujii et al., 2020). However, direct measurements of nutrient concentrations and stocks in wood and fine roots are scarce in tropical forests (Fe ldpausch et al., 2004; Li et al., 2010; Heineman et al., 2016).

 While the cerradão showed higher total NPP and higher nutrient demand on all elements for canopy production, cerrado vegetation was less efficient in producing wood, demanding a higher amount of most nutrients (particularly P and K)

290 despite showing similar woody NPP (Fig. 3). Differences in fine-root nutrient demand between cerrado and cerradão varied by nutrient. The cerrado plot showed similar fine-root NPP to the cerradão and demanded a similar amount of N, Ca and Mg. However, cerrado species demanded more P and K to the fine roots, suggesting that generally cerrado species were also less efficient in building fine roots. Fine roots are short-lived and are the primary pathway for water and nutrient uptake by plants (Eissenstat and Yanai, 1997; McCormack et al., 2012). Hence, their high turnover is important for ecosystem-level processes

295 such as nutrient cycling, resource capture, and global biogeochemistry (Hendricks et al., 1993). Globally, savannas account for ~20% of total fine root biomass (Jackson et al., 1996) , and specifically in the Brazilian savanna, increasing fine roots biomass reflected in a higher uptake of limiting nutrients in soil (Loiola et al., 2016), since fine roots are the principal organs responsible for cation uptake from the soil. In addition, it was also shown that fine roots in deep layers of the soil are essential to overcome the strong water deficit during the dry season in the Cerrado (Oliveira et al., 2005). Therefore, the

300 efficient fine root production in the cerradão vegetation may be an important adaptation to readily absorb water and nutrients during the strongly seasonal rainfall period (February et al., 2013).
## 5 Conclusion

By using the direct relationship between nutrient stoichiometry and NPP for different biomass components, we showed that cerrado vegetation is less efficient in fine root and wood production, requiring more nutrients per unit of NPP compared to the cerradão. Ultimately, we also report the different strategies in P use between cerrado and cerradão that may reflect savanna vegetation adaptation to the indirect effects of fire in the soil biota, which is a recurrent disturbance that increases the risk of losing P. With the increasing fire frequency associated with extreme drought events due to climate change, cerradão vegetation, which strongly rely on rapid nutrient cycling, may suffer further P limitation.

## 6 Author contribution

MCS, IO and YM conceived the idea, questions and hypotheses; IO and YM secured funding for research; MCS, IO, BSM, BHMJr and YM developed study design and methodology; MCS, RF and KSP collected the data; MCS and SR analysed the data; MCS led the writing of the manuscript, and all authors critically contributed to the drafts and gave final approval for publication.

## 7 Competing interests

The authors declare that they have no conflict of interest.

## 8 Acknowledgments

We thank NERC (Natural Environmental Research Council) BIO-RED (#ACR00460) for funding this research. We also thank Camila Borges Silva, Halina Jancoski, Edmar Almeida and Igor Araújo for the valuable help during fieldwork. MCS is supported by a postdoctoral fellowship from CAPES (Coordenação de Aperfeiçoamento de Pessoal de Nível Superior), Brazil. YM is supported by the Jackson Foundation.

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
