# Peer review of "Contrasting strategies of nutrient demand and use between savanna and forest ecosystems in a Neotropical transition zone"

_Biogeosciences, 2022_

## Referee Report (RR1)

Introduction has been improved following previous comments and suggestions, as well as hypotheses, which are now clarified. Materials and methods have been clarified. I would only homogenize the terminology referring to replicates. For example, what is area referred to line in 135, subplot or plot? Results and discussion have also been improved.

---

## Author Response (AR2)

**In general, referee #1 asked for more details and information regarding concepts in the introduction, details that were missing or unclear in the methods and statistical analysis. We believe all major and minor comments were addressed – we are providing all required information in the text and added 2 new supplementary tables with statistical results from the analyses. Also, there were indeed some mismatches between the text and figures and we acknowledge both of the reviewer for noticing it. Figures were redrawn and all minor recommendations were accepted. Please find our point by point answer to the referee #1 comments.**

Review #1

Introduction needs some information. Particularly in relation to introduce important concepts for the better understanding of this study such as nutrient concentration, nutrient resorption, nutrient demand vs nutrient uptake, and then nutrient uptake efficiency vs nutrient use efficiency. Furthermore, the hypotheses will should clarify whether the study focuses on the species scale, on the community scale or both.

**Reply: New information was added and hypotheses' scale were better clarified. Specifically, we now define: "By definition, nutrient use efficiency is the amount of production per nutrient unit (Chapin, 1980) and can be estimated as the ratio of. NPP per unit of nutrient demand (Bridgham et al., 1995). Nutrient demand is the sum of nutrient accumulated in above and belowground biomass and nutrient returns to the soil via litterfall including resorption efficiency, while nutrient uptake excludes nutrient resorption efficiency. Nutrient resorption is defined as the process from which plants withdraw nutrients from senescent leaves prior to leaf abscission, and its efficiency is calculated as the proportional resorption from green to senesced leaves (Killingbeck 1996)."**

2.- Materials and methods need some clarification. Particularly in relation to the experimental design of field sampling. Information is needed on how many replicates were sampled. Are there only two replicates per vegetation type (cerrado and cerradão), and they are referred to as plots in the manuscript? Where were soil samples collected, under the tree canopy or outside? How was species abundance measured, which method was followed? Could you describe species abundance and basal area? All species are found in the all replicates? Also, all sampled species are trees? I recommend adding this information in table S1, and this table add on the manuscript. Information is also needed on how you have measured net primary productivity. It is really important parameter in this study, and there is very little information in the methods. Also, were senescent leaves collected from the same individuals collected previously in January 2008? Information on analytical techniques of soil data is also needed.

**Reply: We understand the reviewer concern with the study scale, especially because we are scaling up from species to community and ecosystem functioning. We also only have one replicate of each site, i.e., for NPP calculations the sample unit is one. For that reason, we had to use error propagation techniques throughout the subplots and community weighted means to represent the community (as you already noted in your comment below).**

We clarified now species selection criteria and inserted Table 1 (old Table S2) in the main document, with IVI and relative dominance values for each species. We also included all information required on this regard (5 individuals were used and the same individuals were sampled in both seasons): "Calculation of relative dominance was based on previous census collected in the area, by dividing the species dominance (i.e., total basal area of the species) and the sum of the dominance of all species multiplying by 100. The importance value index (IVI) was calculated as the sum of the relative frequency, the relative density and the relative dominance of each species within the community. We choose only adult trees with at least 5cm in diameter at breast high (dbh) in the cerrado and 10 cm dbh in the cerradão. For each species, 5 individuals were chosen, from which we collected samples…"

For soil samples, we noted that we included only data from two soil depths – the text was changed accordingly with more details added as well: "Within each 1ha plot, 20 subplots were delimited, from where soil samples at 0-10 and 10-20 cm depth were collected in all 4 corners and in the centre of each subplot, totalling 200 samples for each vegetation type. Soil chemistry data were analysed according to EMBRAPA procedure (EMBRAPA, 1997) and were provided by ForestPlots database Lopez-Gonzalez et al. 2011 (Table S1)."

Primary productivity measurements were briefly described – more detailed description of NPP calculations can be found in associated published literature from our group. It now reads: "Here we quantified the major components of NPP, including the canopy (leaves, twigs and reproductive parts), wood (stem, coarse roots and branches), and fine roots during 2014 to 2016. Data were collected following GEM protocols (Malhi et al. 2021) and methods are described in detail in Mathews et al. (2014) and Malhi et al. (2015). Briefly, for the canopy NPP component estimation, litter traps sampled biweekly together with monthly canopy leaf area index were used. For wood component estimation, annual census and dendrometers measuring growth rates were converted into woody biomass production. Fine root production was measured with ingrowth cores installed and sampled every three months.

3.-Information on some statistical methods needs to be improved. In particular, the reason behind the use of community weighted mean to scale up species values to community for nutrient concentrations, which affects nutrient demand, nutrient use, and nutrient use efficiency and nutrient uptake efficiency parameters at community scale. Species selection would produce a strong bias in the community value, especially when target species belong to different families. Could you justify species selection in the methods, indicating for example their abundance in plots. Furthermore, the use of community weighted mean is only suitable for use with many replicates to avoid Type I error, or to include random effects on the models. For this reason, more information is needed on the statistical methods. Did you include any random factors on the MANOVAs and ANOVAs? What are the variables, the fixed terms and the random terms? And what R function did you use?

Reply: Information were added. We now specify how calculations of CWM were performed as well as dependent and independent variables used in MANOVA. Species

selection criteria were already explained in the section below – see coment #2. We used function 'manova' from 'stats' package, however we do not see the necessity to cite base R packages, or functions.

"To up-scaled species value for the whole communities, we calculated the community-weighted mean (CWM) for each organ using species relative dominance to weight the nutrient concentration (Muscarella & Uriarte, 2016). To compare community-weighted nutrient concentration average means and resorption efficiencies between the two vegetation types and between different organs, we performed a two-way multivariate analysis of variance (MANOVA) followed by univariate analysis of variances (ANOVA's) and Tukey HSD post-hoc test. The independent variables were site and plant organ while the dependent variables were the different nutrient concentrations (N, P, K, Ca, Mg) or resorption. Data normality and homogeneity of variances assumptions were previously checked with Shapiro-Wilk multivariate normality test using package 'mvnormtest' (Jarek and Jarek, 2009) and Levene test using package 'car' (Fox et al., 2012), respectively. All statistical analyses were performed in R software version 4.0.1 (R Core Team, 2019)."

Discussion section would be clearer if separate paragraphs were used to discuss each hypothesis, indicating the key results of this study. In this sense, the authors dedicate the first and the second paragraphs to discuss a higher P content in wood by Cerrado species than Cerradão species as a key result when they did not report any statistically significant test value in the results for sapwood and heartwood (line 178, Fig1, Fig S1), as they did for inner bark. Could they justify this or report a test value in the results?
Reply: We re-ordered Discussion section to follow the three suggested hypothesis. Indeed, for some unknown reason the asterisks were not displayed in Fig 1. Differences are now shown in the figure and tests were acknowledge in the text. At the species scale, there was significant difference between cerrado and cerradão species for P content in wood. At the community scale, even though there was no difference between CWM nutrient concentration, there was also remarkable difference in P demand for the wood component and therefore we wanted to highlight this finding in the discussion.

-Throughout the manuscript try to homogenize concepts as plots, area or sites, and to differentiate between species scale or community scale.

Reply: Done.

-Species name would be in italic format in the text, such as *Hirtella glandulosa* and *Emmotum nitens* in line 117.

Reply: Done.

-Plant nutrient concentrations would be in mg/g instead of %.

Reply: Done.

- You should clarify the statements of results and discussion in lines 229-230 and 278-280, because they can be misinterpreted. Ca uptake should always match demand and never resorption because the differences between nutrient demand vs nutrient uptake, and nutrient uptake efficiency vs nutrient use efficiency is based on use or not the nutrient resorption efficiency, which for Ca is zero.

**Reply: We agree. Our intention was to compare between sites (cerrado vs. cerradão). We rephrased both statements only mentioning Mg differences, since Ca was used in the calculation for resorption metrics, to avoid confusion.**

-Are there significant differences between sites on N uptake efficiencies? On the figure 4 is indicated, but not on the text (line 224).

**Reply: There is not (P-value is 0.078). Figure asterisks were placed incorrectly, and we are very sorry about that. We double checked this issue in new figures.**

-I recommend modifying Figure 4 and deleting the last row, because it is a repeat of Figure 3. Also, could you please provide test value for the Nutrient Use efficiency of P, because it does not seem significant in Figure 4, as well as for the nutrient uptake efficiency of K?

**Reply: We added supplementary table with results from z-tests (transformed p-values). Deleted the last row of Figure 4, as it has redundant information. Asterisks are now displayed correctly.**

- I recommend reducing the importance of statements related to fine root production, because the sampling carried out is not accurate and other non-target species, such as grasses, could be measured.

**Reply: Done. It now reads: "However, cerrado species demanded more P and K to the fine roots, suggesting that generally cerrado species require more nutrient for fine roots production. Fine roots are important component of the biomass, reflecting in a higher uptake of limiting nutrients in soil (Loiola et al., 2016), and are essential to overcome the strong water deficit during the dry season in the Cerrado (Oliveira et al., 2005). Therefore, the efficient fine root production in the cerradão vegetation may be an important adaptation to readily absorb water and nutrients during the strongly seasonal rainfall period (February et al., 2013). However, since our sampling strategy may have measured non-target species, such as grasses, our results could be biased."**

- I recommend to avoid any reference to figures or tables in the discussion, because they should be indicated on the results.

**Reply: Done. All reference to tables and figures in the discussion were removed.**

**Reviewer #2 suggested using different terms for nutrient uptake and use efficiency, which could be a bit misleading and suggested some missing references and corrected some wrong citations. We addressed all comments, changed the terms following reviewer's suggestion and revised all references. Please find our point by point answer to the reviewer #2 comments.**

Reviwer #2

Nutrient use and uptake efficiencies: These two terms a bit confused me as I thought that the uptake efficiency indicates the efficiency of nutrient uptake per unit uptake cost (or unit carbon or something like that). However, the uptake efficiency was calculated as the ratio of NPP to unit mass of the nutrient that was taken up from soils. Perhaps, 'nutrient-use efficiency (uptake basis)' or 'NuUEuptake' might be a more suitable term for example. Similarly, the use efficiency could be described as 'nutrient use efficiency (demand basis)' 'NuUEdemand'. Also add the definition of nutrient use efficiency in the abstract.

**Reply: Done. We added definition for nutrient use efficiency in the abstract (, i.e., the amount of production per nutrient unit). We also added definition of all terms used (nutrient efficiency, nutrient demand and uptake, nutrient resorption) in the first paragraph of the introduction. We used the suggested term for nutrient use efficiency uptake or demand base.**

There were missing or errors in citations. I listed them in the specific comments. Please carefully double-check the literatures.

L 15 I recommend the authors delete 'for the first time'. There is a study that simulated vegetation-level nutrient-use efficiencies and flux by coupling NPP with stoichiometry.
**Reply: Done.**

Wang, Y., Ciais, P., Goll, D., Huang, Y., Luo, Y., Wang, Y.-P., … Zechmeister- Boltenstern, S. (2018). GOLUM-CNP v1.0: A data-driven modeling of carbon, nitrogen and phosphorus cycles in major terrestrial biomes. Geoscientific Model Development, 11, 3903–3928. https://doi. org/10.5194/gmd-11-3903-2018

L 24-26 I recommend the authors rewrite or delete this statement. I did not get how the authors evaluated the efficiencies of fine root and wood production. Did the authors calculate nutrient use efficiency in the production of fine roots or wood?
**Reply: We understand the confusion we may have caused using the term efficiency to describe nutrient allocation in relation to biomass production. We re-worded here and throughout the text, avoiding using "efficiency", but clearly stating whether there was more or less nutrient allocated to a given biomass. It now reads: "The proportional difference in nutrient allocation to the different biomass components suggesting cerrado species allocate less nutrient to a given fine root biomass, but more nutrient to a given wood biomass."**

L 27 how did the authors know the P and K limitation in the forests? I need the evidence that the forests are considered under P and K limitation. For example, P- or K-resorption efficiency was higher than global average, etc.

**Reply: We meant cerradão species were more limited in P and K than cerrado species – sentence was reworded. "Our findings suggest that cerradão species are more limited in P and K than cerrado species, inducing a higher resorption to compensate for low uptake."**

L 28 I am not sure if this is a trade-off or not. I think that trees can increase N uptake and N-use efficiency simultaneously.

**Reply: We agree. Sentence now reads:**
**"This difference in nutrient dynamics explains how similar soils and the same climate dominated by savanna vegetation can also support forest-like formations."**

L 29-30 I thought that this simply means that Ca and Mg were little resorbed before leaf fall.

**Reply: We deleted this sentence from the abstract to avoid confusion. We were referring to a comparison between sites (cerrado vs. cerradão) but we understand it was misleading.**

L 30 'species composition' came out of nowhere. It would be good to clarify why species composition can be the major factor.

**Reply: We now added the information that the communities were composed by different species in the beginning of the abstract: Here, we describe different nutrient use and allocation strategies in Neotropical savanna (cerrado) and transitional forest (cerradão) tree communities composed by different species, report leaf nutrient resorption and calculate ecosystem-level nutrient use efficiency."**

INTRODUCTION: the introduction was well edited.
L 73-76 it would be good to add references to these sentences.
**Reply: Done – we added Vergutz et al. (2012).**

METHOD:
L 143-144 As much as I remember, MLCF in Vergutz et al. 2012 is the ratio of green-leaf mass to senescent leaf mass but not Ca. Please double-check.
**Reply: Yes, you are correct. We removed this reference and added Vitousek and Sanford (1986)**
**Vitousek, P. and Sanford R. Nutrient cycling in moist tropical forest. Ann Rev Ecol Syst 17: 137-167. https://doi.org/10.1146/annurev.es.17.110186.001033, 1986.**

L 144-145 I would recommend the authors provide the equation to calculate community weighted manes.
**Reply: This was deleted from this paragraph and detailed in the statistical analysis subsection.**

L 149-150 Add brief explanations for the NPP measurement. I was wondering if the NPP was estimated by litterfall monitoring and tree census.
**Reply: As requested, this information is now provided: "Data were collected following GEM protocols (Malhi et al. 2021) and methods are described in detail in Mathews et al.**

**(2014) and Malhi et al. (2015). Briefly, for the canopy NPP component estimation, litter traps sampled biweekly together with monthly canopy leaf area index were used. For wood component estimation, annual census and dendrometers measuring growth rates were converted into woody biomass production. Fine root production was measured with ingrowth cores installed and sampled every three months."**

L 157-158 As I mentioned in the major concerns, the nutrient uptake efficiency might be a bit misleading.
**Reply: We changed to nutrient use efficiency (uptake basis) following your suggestion**

L 162 I think this sentence includes typos
**Reply: Sentence was changed.**

RESULTS:
L 240-242 Please make where this statement came clear. Maybe, Tsujii et al. 2020?
Tsujii Y, Aiba S-I, Kitayama K. Phosphorus allocation to and resorption from leaves regulate the residence time of phosphorus in above-ground forest biomass on Mount Kinabalu, Borneo. Funct Ecol. 2020;34: 1702–1712.
https://doi.org/10.1111/1365-2435.13574
**Reply: Reference was added.**

L 242-245 Which results support this statement?
L 245-246 Aoyagi & Kitayama (2016) is a good reference for this statement but not for the following statement (L 247-248).
L 248 Aoyagi & Kitayama (2016) might not focus on P residence time. Please double check this reference.
**Reply: Yes, we are sorry about that. We deleted this reference and left only Tsujii et al. 2020 to support the statement. We included Aoyagi & Kitayama (2016) as a reference in the previous statement on the mechanism of allocating P to the canopy to maintain higher photosynthetic rates.**

L 250 P content in wood may be also affected by reproductive status, such as masting. For example,
Ichie, T., & Nakagawa, M. (2013). Dynamics of mineral nutrient storage for mast reproduction in the tropical emergent tree Dryobalanops aromatica. Ecological Research, 28(2), 151–158. Retrieved from https://doi.org/10.1007/s11284-011-0836-1
**Reply: That is an interesting reference but we decided to not include it since we were not controlling for reproductive status of the vegetation.**

L 284-258 Please carefully check the citations. As much as I remember, Aoyagi & Kitayama (2016) did not estimate P residence time. Tsujii et al. (2020) estimated P residence time in above-ground forest biomass (canopy + wood). Gleason et al. estimated P residence time in canopy, but also estimated P-use efficiency at the above-ground biomass level (i.e. including canopy and wood). In addition to these papers, Paoli et al. (2005) estimated P residence time in canopy.
Paoli, G. D., Curran, L. M., & Zak, D. R. (2005). Phosphorus efficiency of Bornean rain forest productivity: Evidence against the unimodal efficiency hypothesis. Ecology, 86(6),

1548–1561. Retrieved from https://doi.org/10.1890/04-1126

**Reply: Thank you for noticing that. Indeed, we wanted to refer to aboveground biomass instead of canopy. We also included the reference suggested.**

L 285-286 The following papers analysed nutrient concentrations and estimated nutrient stocks in wood and/or fine roots for tropical trees.

Hughes, R. F., Kauffman, J. B., & Jaramillo, V. J. (1999). Biomass, Carbon, and Nutrient Dynamics of Secondary Forests in a Humid Tropical Region of Mexico. Ecology, 80(6), 1892. Retrieved from https://doi.org/10.2307/176667

Imai, N., Kitayama, K., & Titin, J. (2010). Distribution of phosphorus in an above-tobelow-ground profile in a Bornean tropical rain forest. Journal of Tropical Ecology, 26(06), 627–636. Retrieved from https://doi.org/10.1017/S0266467410000350

Johnson, C. M., Vieira, I. C. ., Zarin, D. J., Frizano, J., & Johnson, A. H. (2001). Carbon and nutrient storage in primary and secondary forests in eastern Amazônia. Forest Ecology and Management, 147(2–3), 245–252. Retrieved from https://doi.org/10.1016/S0378-1127(00)00466-7

Kauffman, J. B., Cummings, D. L., Ward, D. E., & Babbitt, R. (1995). Fire in the Brazilian Amazon: 1. Biomass, nutrient pools, and losses in slashed primary forests. Oecologia, 104(4), 397–408. Retrieved from https://doi.org/10.1007/BF00341336

Tsujii Y, Aiba S-I, Kitayama K. Phosphorus allocation to and resorption from leaves regulate the residence time of phosphorus in above-ground forest biomass on Mount Kinabalu, Borneo. Funct Ecol. 2020;34: 1702–1712. https://doi.org/10.1111/1365-2435.13574

**Reply: Citations were added accordingly.**

CONCLUSION:
L 303-305 It might be good to say 'the cerrado vegetation allocated more nutrient to root and wood' rather than say 'less efficient in their production'.

**Reply: We agree with this suggestion and changed accordingly.**

Tables & Figures:
Figure 1 I did not find asterisks.

**Reply: Indeed, for some unknown reason the asterisks were not displayed in Fig 1. Differences are now shown in the figure and tests were acknowledge in the text.**

Associate Editor asked us to provide further justification and explain the limitations of having only 1 site per vegetation type.

The two sites are part of the Global Ecosystem Monitoring (GEM) network, where the standard site is a 1 hectare square (100 x 100 m), which "is considered an adequate size to sample a range of trees (typically 500-800 trees > 10 cm dbh) and not be overly influenced by individual tree gap dynamics, while also being a tractable area to sample at high frequency" (Malhi et al. 2021).

Obviously, GEM protocol has some limitations, such as the associated uncertainties to the multiple measurements/estimates that compose ecosystem carbon cycle. However, to overpass this limitation, each of these uncertainties are accounted for by rigorous error propagation during summation.

Finally, many different studies under the same protocol have largely contributed to our understanding in carbon fluxes and stocks in different system, especially in the tropics (Aragão et al. 2009; Doughty et al. 2014a, 2014b, 2015a, 2015b; Girardin et al. 2016; Kho et al. 2013; Malhi et al. 2017; Moore et al. 2018, among others).

We added to the main text: "However, it is important to note that our results are based on two sites, and there may be potential misinterpretation due to any particularity of these studied sites. Even though we examined the only Cerrado and Cerradão established sites with intensive monthly data collection and monitoring, our findings would obviously benefit from further testing with more savanna and transition forest sites."

Aragão, L., Y. Malhi, D. Metcalfe, J. Silva-Espejo, E. Jiménez, D. Navarrete, S. Almeida, A. Costa, N. Salinas, and O. Phillips. 2009. Above-and below-ground net primary productivity across ten Amazonian forests on contrasting soils. Biogeosciences 6:2759-2778.

Doughty, C. E., Y. Malhi, A. Araujo-Murakami, D. B. Metcalfe, J. E. Silva-Espejo, L. Arroyo, J. P. Heredia, E. Pardo-Toledo, L. M. Mendizabal, V. D. Rojas-Landivar, M. Vega-Martinez, M. Flores-Valencia, R. Sibler-Rivero, L. Moreno-Vare, L. J. Viscarra, T. Chuviru-Castro, M. Osinaga-Becerra, and R. Ledezma. 2014a. Allocation trade-offs dominate the response of tropical forest growth to seasonal and interannual drought. Ecology 95:2192-2201.

Doughty, C. E., D. B. Metcalfe, M. C. da Costa, A. A. R. de Oliveira, G. F. C. Neto, J. A. Silva, L. Aragao, S. S. Almeida, C. A. Quesada, C. A. J. Girardin, K. Halladay, A. C. L.

da Costa, and Y. Malhi. 2014b. The production, allocation and cycling of carbon in a forest on fertile terra preta soil in eastern Amazonia compared with a forest on adjacent infertile soil. Plant Ecology & Diversity 7:41-53.

Doughty, C. E., D. B. Metcalfe, C. A. J. Girardin, F. F. Amezquita, D. G. Cabrera, W. H. Huasco, J. E. Silva-Espejo, A. Araujo-Murakami, M. C. da Costa, W. Rocha, T. R. Feldpausch, A. L. M. Mendoza, A. C. L. da Costa, P. Meir, O. L. Phillips, 1171 and Y. Malhi. 2015a. Drought impact on forest carbon dynamics and fluxes in Amazonia. Nature 519:78-U140.

Doughty, C. E., D. B. Metcalfe, C. A. J. Girardin, F. F. Amezquita, L. Durand, W. H. Huasco, J. E. Silva-Espejo, A. Araujo-Murakami, M. C. da Costa, A. C. L. da Costa, W. Rocha, P. Meir, D. Galbraith, and Y. Malhi. 2015b. Source and sink carbon dynamics and carbon allocation in the Amazon basin. Global Biogeochemical Cycles 29:645-655.

1181 Biogeosciences 122:2952-2965.

Girardin, C. A. J., Y. Malhi, C. E. Doughty, D. B. Metcalfe, P. Meir, J. del Aguila-Pasquel, A. Araujo-Murakami, A. C. L. da Costa, J. E. Silva-Espejo, F. F. Amezquita, and L. Rowland. 2016. Seasonal trends of Amazonian rainforest phenology, net primary productivity, and carbon allocation. Global Biogeochemical Cycles 30:700-715.

Kho, L. K., Y. Malhi, and S. K. S. Tan. 2013. Annual budget and seasonal variation of aboveground and belowground net primary productivity in a lowland dipterocarp forest in Borneo. Journal of Geophysical Research-Biogeosciences 118:1282-1296.

Malhi, Y., C. A. J. Girardin, G. R. Goldsmith, C. E. Doughty, N. Salinas, D. B. Metcalfe, W. H. Huasco, J. E. Silva-Espejo, J. del Aguilla-Pasquell, F. F. Amezquita, L. Aragao, R. Guerrieri, F. Y. Ishida, N. H. A. Bahar, W. Farfan-Rios, O. L. Phillips, P. Meir, and M. Silman. 2017. The variation of productivity and its allocation along a tropical elevation gradient: a whole carbon budget perspective. New Phytologist 214:1019-1032.

Moore, S., S. Adu-Bredu, A. Duah-Gyamfi, S. D. Addo-Danso, F. Ibrahim, A. T. Mbou, A. de Grandcourt, R. Valentini, G. Nicolini, G. Djagbletey, K. Owusu-Afriyie, A. Gvozdevaite, I. Oliveras, M. C. Ruiz-Jaen, and Y. Malhi. 2018. Forest biomass, productivity and carbon cycling along a rainfall gradient in West Africa. Global Change Biology 24:E496-E510.